# Effect of Hot Isostatic Pressing Process Parameters on Properties and Fracture Behavior of Tungsten Alloy Powders and Sintered Bars

**DOI:** 10.3390/ma15238647

**Published:** 2022-12-04

**Authors:** Biao Hu, Gaoshen Cai

**Affiliations:** School of Mechanical Engineering, Zhejiang Sci-Tech University, Hangzhou 310018, China

**Keywords:** hot isostatic pressing, tungsten alloy, mechanical properties, fracture behavior

## Abstract

In order to investigate the effect of hot isostatic pressing (HIP) process parameters on the properties and fracture behavior of tungsten alloy, HIP experiments with different process parameters were carried out, and the relative density, Rockwell hardness, tensile properties, and tensile fracture behavior were analyzed. The results show that after HIP, the tungsten alloy samples obtained further densification, higher relative density and hardness, and lower dispersity. At 1300 °C and 140 MPa, the sintered bar achieved excellent mechanical properties: yield strength increased by 16.5%, tensile strength increased by 16.1%, and fracture strength increased by 85.3%. Comparing the two processes, the mechanical properties of tungsten alloy powders formed directly via HIP were not as good as those of the sintered bars. In addition, after HIP, the fracture mode of the tungsten alloy sintered bar samples was mainly ductile tear, and that of the tungsten alloy powder samples was mainly a full brittle fracture.

## 1. Introduction

Tungsten heavy alloy is an alloy material composed of tungsten as the base material (with a tungsten content of 85%~99%) and a small amount of metal binders, such as nickel (Ni), copper (Cu), iron (Fe), cobalt (Co), molybdenum (Mo), and chromium (Cr) [1]. The density of tungsten heavy alloy can reach 19.3 g/cm^3^ [2]. Tungsten alloy is a prominent metal among the refractory metals due to its high melting point, high density, excellent high temperature strength, good thermal conductivity, and low coefficient of thermal expansion [3,4,5]. The advantages of these properties make it widely used in aviation, aerospace, and military applications [6,7].

However, tungsten alloy is a material that is difficult to process [8]. Therefore, it is necessary to explore new processing methods to form tungsten alloy materials. Hot isostatic pressing, as a promising advanced manufacturing technology, enables rapid densification between powder particles inside the blanks by imposing high temperature and high pressure on them [9,10]. It not only can strengthen the powder materials in the forming process [11] but can densify and strengthen the sintered materials that have been formed [12], which can combine the forming process and strengthening process. Therefore, HIP is widely applied to produce high-level components.

It is reported that HIP can improve the properties of tungsten alloy [13,14,15]. Rabouhi et al. [16] studied the microstructural and mechanical characterization of tungsten alloys fabricated by liquid phase sintering and HIP. They found an obvious improvement in mechanical properties by using HIP compared to liquid phase sintering. Munoz et al. [17] investigated the microstructural and mechanical characteristics of tungsten alloy materials processed by HIP. Wang et al. [18] found that the density of the tungsten alloy samples was increased to 99.4% after treatment by HIP. However, there was little effect of HIP process parameters on the properties and fracture behavior of tungsten alloy.

In this study, to investigate the effect of HIP temperature and pressure on the properties and fracture behavior of 93W-Ni-Fe heavy alloy powders and sintered bars, HIP experiments with different process parameters were carried out. The relative density, Rockwell hardness, tensile properties, and tensile fracture behavior of tungsten alloy materials fabricated by HIP were analyzed.

## 2. Materials and Methods

### 2.1. Raw Materials

The material used in this study was 93W-Ni-Fe heavy alloy. Tungsten powders, iron powders, and nickel powders with 99.98% purity were used as raw materials. The main chemical composition of the 93W-Ni-Fe heavy alloy is shown in Table 1. First, each raw powder was weighed according to the mass fraction in Table 1. Then, the raw powder was placed in a planetary ball mill for 8 h mechanical ball grinding, and argon gas was used for oxidation protection during ball grinding. The raw mixed powders were repressed by mold forming; then, liquid-phase sintering was carried out in a molybdenum wire furnace. Hydrogen gas was used as protection gas in the liquid-phase sintering process. The liquid-phase sintering process parameters are shown in Figure 1. An SEM (HITACHI S-4800, Tokyo, Japan) image of the tungsten alloy powders is shown in Figure 2a. It can be seen from Figure 2a that there are mass pores between the powders. The relative density of the tungsten alloy powders reached 0.53. Figure 2b shows the morphology of sintered bars. The relative density of the bar reached 0.923.

### 2.2. HIP Process

A schematic of the HIP process is shown in Figure 3. A total of 45 carbon steel capsules with a three dimension (wall thickness 2 mm, inner diameter 20 mm, and length 200 mm) were fabricated by the machining process (Figure 3a). After mechanical vibration of loading powders, the relative density of powders in the capsule reached about 0.6 (Figure 3b). The capsules were degassed by a molecular pump until the vacuum degree was 9 × 10^−5^ Pa (Figure 3c). Subsequently, tungsten argon arc welding was used to weld and seal the capsule (Figure 3d). Because of its surface compactness, the sintered bars can be processed directly by HIP without the requirement of capsule sealing. Finally, the sealed capsules and sintered bars were deposited into the RD-450 type HIP equipment (Aetna technology company, Beijing, China) for sintering and densification (Figure 3e). The main parameters of the equipment are listed in Table 2. During the HIP process, the temperature and pressure increased to a designated value within 2 h and then were maintained at the same value for 4 h. Finally, the temperature and pressure gradually dropped to room temperature and pressure within 2 h. The process parameters of HIP are shown in Table 3, and the temperature and pressure loading curves are shown in Figure 4.

### 2.3. Characterization

In order to explore the densification effect of HIP on tungsten alloy sintered bars and powder materials, the density of samples processed by HIP was measured by an electronic balance (AR224CN, Ohaus, Pine Brook, NJ, USA) based on the Archimedes drainage method. To ensure accuracy, 5 small blocks were taken from different parts of each sample for testing, and the average density was obtained. The theoretical density of the tungsten alloys is 18.595 g/cm^3^ according to the calculation of the chemical composition. The initial density of powders is the ratio of the mass of powders inside the capsule to the volume of the capsule.

The Rockwell hardness of tungsten alloy samples before and after HIP was measured according to the GB/T2301–2004 Standard of Rockwell hardness testing for metallic materials. An energy dispersive X-ray spectroscopy (EDX-720, Kyoto, Japan) was employed to analyze the chemical composition and element distribution of the tungsten alloy samples. The tensile specimens were tested using a tensile test machine (FPZ100, Germany) based on GB/T228.1-2010 metallic material tensile testing under the condition of the loading speed (1.5 mm/min). In addition, the tensile fracture morphologies were observed by a scanning election microscope (SEM, HITACHI S-4800, Tokyo, Japan).

## 3. Results and Discussion

### 3.1. Relative Density

The average relative density of tungsten alloy before and after HIP is shown in Figure 5. After HIP treatment, the sintered bars were almost completely densified. At 1300 °C and 140 MPa, the relative density of the sintered bar was greatly improved to 99.66 %. For powder materials, however, the density of solid phase sintering was not optimal under the two process parameters. With the increase in temperature, further densification of the powder materials was achieved. It can be seen from Figure 5 that both the temperature and pressure of HIP can increase the density of tungsten alloy. However, temperature played a greater role in the material densification process than pressure. When the temperature increased by 100 °C, the relative density increased by 2.3%. It can be predicted that if the temperature is further increased, the powder materials will gradually experience full densification.

### 3.2. Rockwell Hardness

Figure 6 shows the comparison of the Rockwell hardness of tungsten alloy before and after HIP. It can be found that the hardness was greatly improved after further densification. The hardness of the material depends on the internal defects of the material. Due to the melting point of W being obviously different from that of Fe and Ni, the flow of Fe and Ni is obvious after liquid phase sintering, resulting in a shrinkage cavity and porosity [20]. Through the action of high temperature and high pressure, the internal shrinkage of the material was further realized, and the dispersion was significantly reduced.

It can be seen from Figure 6 that the hardness of the material was significantly improved after HIP, and the dispersion of the hardness of the material before HIP was obviously reduced. This shows that HIP can improve the composition segregation and eliminate defects such as pores. In the comparison experiment of three groups of sintered bars, the hardness was the largest at 1300 °C and 140 MPa, and the corresponding relative density was also the largest. For tungsten alloy powders, the increase of HIP temperature can markedly increase the density of the material, thereby improving the hardness of the material. But under the same process parameters of HIP, its hardness value was significantly less than that of sintered bars.

### 3.3. EDX Analysis

In the solid-phase sintering process of tungsten alloy, Ni and Fe with high solubility can form a complete solid solution, while W has good solubility in Ni and Fe, which can form the γ-(Ni, Fe, W) phase. The process is divided into two main stages. First, due to the low melting point and high solubility of Ni and Fe, the mutual diffusion of Ni and Fe will result in the formation of the γ-(Ni–Fe) phase. Then, as the temperature increases, W begins to diffuse into the γ-(Ni–Fe) phase, thus achieving the alloying process.

Figure 7 and Table 4 show the chemical composition and element distribution of the sintered bar strengthened by HIP. As can be seen from Figure 7a, after HIP treatment at 1200 °C and 140 MPa, Ni and Fe account for 48.27% and 25.64% of the bonding phase of the tungsten alloy. The γ-(Ni–Fe) phase of the solid solution was basically homogenized by diffusion. Meanwhile, the diffusion of W occurred, and the diffusion of W to the bonding phase reached 18.87%. Under these conditions, the alloying process was basically complete. The density of the sinter bar was increased from the original 92.3% to 98.8%. It can be seen from Figure 7b that when the temperature increased from 1200 °C to 1300 °C under the pressure of 140 MPa, the solubility of W in the γ-(Ni–Fe) phase further increased, but the increasing degree was small, only from 18.87% to 20.91%. The chemical composition of tungsten grains at 1200 °C and 140 MPa is shown in Figure 7c. Comparing Figure 7b,c, it can be seen that compared with the high solubility of W in Ni and Fe, the solubility of Ni and Fe in the W phase is extremely small, accounting for 0.27% and 0.2%, respectively.

In addition, it can be seen from Figure 7c and Table 4 that during the densification process of tungsten alloy, self-diffusion occurs in the W phase. The diffusion of W atoms is closely related to its diffusion activation energy and temperature. The higher the temperature, the more obvious the diffusion degree between W atoms. Since the tungsten phase belongs to the hardening phase, the more obvious the fusion between the tungsten particles, the higher the hardness exhibited. In the comparison experiment, the hardness of tungsten alloy was HRC43.1 at 1300 °C, while the hardness was only HRC41.5 at 1200 °C.

### 3.4. Mechanical Properties

According to the original mechanical properties of sintered bars, the comparison of mechanical properties of materials under different process parameters is shown in Table 5. It can be found from Table 5 that the mechanical properties of tungsten alloy materials were further improved after HIP strengthening compared with those before HIP. Specifically, at 1300 °C and 140 MPa, the sintered bar achieved the best mechanical properties: yield strength increased by 16.5%, tensile strength increased by 16.1%, and fracture strength increased by 85.3%. Different from the deformation strengthening process of tungsten alloy, the HIP strengthening method not only improved the strength but also greatly improved the elongation and percentage reduction of area, which were 46.7% and 43.7%, respectively.

Comparing the three groups of tests of sintered bars, it can be found that increased temperature and pressure could improve the strength, elongation, and reduction of area of the tungsten alloy to a certain extent, but the increase of temperature was more obvious as a strengthening effect.

The properties of tungsten alloy powder materials after HIP were inferior to those of sintered bars, which could be improved by increasing the temperature to a certain extent; however, the optimal temperature and pressure parameters still need to be explored.

It can also be seen from Table 5 that the HIP process can not only improve the mechanical properties of tungsten alloy but also maintain the ductility of the material at a high level.

### 3.5. Tensile Fracture Behaviors

Four main fracture modes of tungsten heavy alloys are shown in Figure 8. Refs. [21,22]: (1) the fracture of the γ phase (A1); (2) transgranular cleavage fracture of tungsten grains (A2); (3) intercrystalline fracture of tungsten grains (A3); (4) dimple fracture of tungsten grains and the γ phase (A4 and A5). After the tensile test, the fracture morphology was observed by a scanning electron microscope. The fracture morphology of the samples under different process parameters are shown in Figure 9.

Figure 9a shows the tensile fracture morphology of tungsten alloy strengthened by HIP at 1200 °C and 140 MPa. The transgranular cleavage fracture area of tungsten grains accounted for about 55%, and most of the bonding phase was ductile tearing, forming a typical dimple fracture, which reflected high tensile strength and good ductility at the micro level. By comparing the tensile fracture morphology of tungsten alloy sintered bars after HIP strengthening (Figure 9a–c), it can be found that under the same pressure, the higher the temperature, the greater the connection strength between tungsten grains and the bonding phase, and the higher the proportion of transgranular cleavage of tungsten grains. In terms of macroscopic mechanical properties, increasing the temperature of HIP could increase the tensile strength at room temperature but decreased the elongation of tungsten alloy.

Figure 9d shows the tensile fracture morphology of powder direct HIP forming at 140 MPa and 1300 °C. It can be seen from Figure 9d that the fracture modes were mainly intercrystalline fracture of tungsten grains and dimple fracture of tungsten grains and the γ phase, which presented a full brittle fracture. In terms of macroscopic mechanical properties, extremely low tensile strength and elongation were observed. In addition, it can be seen from the fracture morphology (Figure 9d) that there are massive pores in the sample. However, the ductility of tungsten alloy is very sensitive to residual porosity. When the porosity exceeds 5%, the tungsten alloy will show obvious embrittlement. When the temperature of HIP was further increased, it was observed from the fracture morphology (Figure 9e) that there was obvious intercrystalline fracture of tungsten grains. This illustrates that the bonding force between particles is enhanced, and the fracture of the γ phase and transgranular cleavage fracture of tungsten grains are increased, which is manifested in the improvement of tensile stress and elongation. Compared with the tungsten alloy sintered bars, the tungsten particles that the sample formed by powder HIP were finer, and the microstructure composition was more uniform. However, because the fracture crack was almost along the W-W interface for the tungsten grains and γ phase, the tensile stress and extensibility of samples formed directly from powders was slightly low.

## 4. Conclusions

Taking 93W-Ni-Fe sintered bars and powder materials as raw materials, the HIP strengthening effect of different materials under different HIP process parameters was compared. The following conclusions could be drawn:(1)The increase of temperature and pressure can improve the strength, elongation, and percentage reduction of area of the materials to a certain extent, but the increase of temperature was more obvious for its strengthening effect. At 1300 °C and 140 MPa, the sintered bar achieved excellent mechanical properties: yield strength increased by 16.5%, tensile strength increased by 16.1%, and fracture strength increased by 85.3%.(2)With the increase of temperature, the diffusivity of W to the γ phase was enhanced, resulting in an increase in the relative density of the sample. At the same time, the self-diffusivity of the W phase was also enhanced, resulting in the improvement of the hardness of the sample.(3)After the HIP, the fracture mode of tungsten alloy sintered bar samples was mainly ductile tear, which reflected high tensile strength and good ductility at the micro level. However, that of the tungsten alloy powder samples was mainly a full brittle fracture, which reflected extremely low tensile strength and elongation at the micro level.(4)Comparing the two processes, the mechanical properties of tungsten alloy powders formed directly via HIP were not as good as those of the sintered bars. However, it is predicted that increasing the temperature and pressure can further improve the mechanical properties. Therefore, in order to further improve the mechanical properties of tungsten alloys, the exploration of optimal process parameters will be carried out in future work.

## Figures and Tables

**Figure 1 materials-15-08647-f001:**
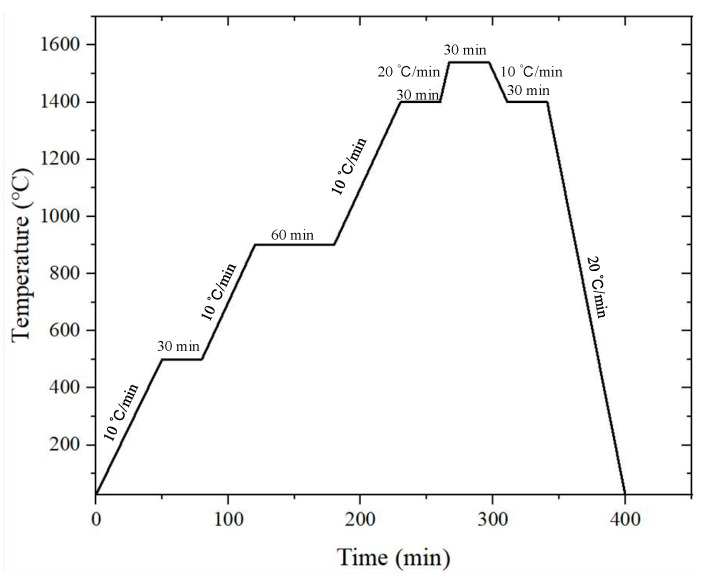
Liquid-phase sintering temperature and time curve.

**Figure 2 materials-15-08647-f002:**
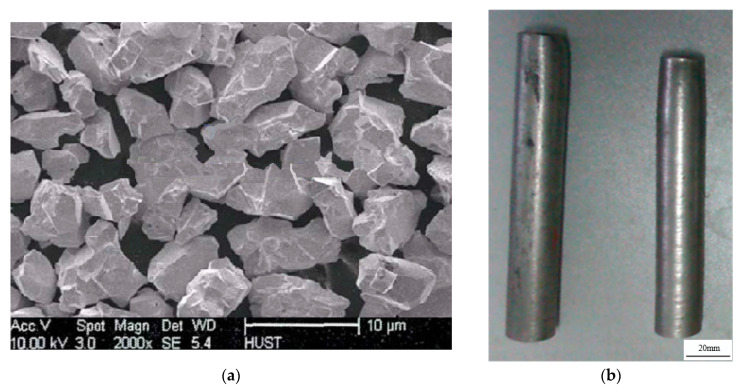
Experimental materials: (**a**) microstructure of tungsten alloy powder materials [19]; (**b**) sintered bars.

**Figure 3 materials-15-08647-f003:**
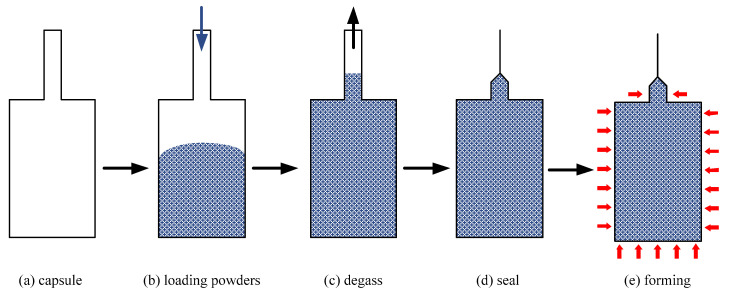
Schematic of HIP process.

**Figure 4 materials-15-08647-f004:**
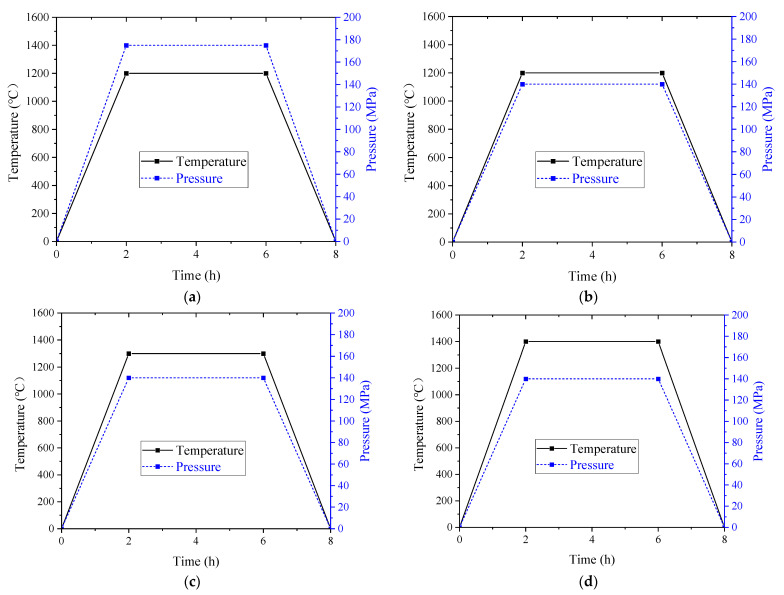
HIP strengthening loading curve of tungsten alloy: (**a**) 1200 °C, 175 MPa; (**b**) 1200 °C, 140 MPa; (**c**) 1300 °C, 140 MPa; (**d**) 1400 °C, 140 MPa.

**Figure 5 materials-15-08647-f005:**
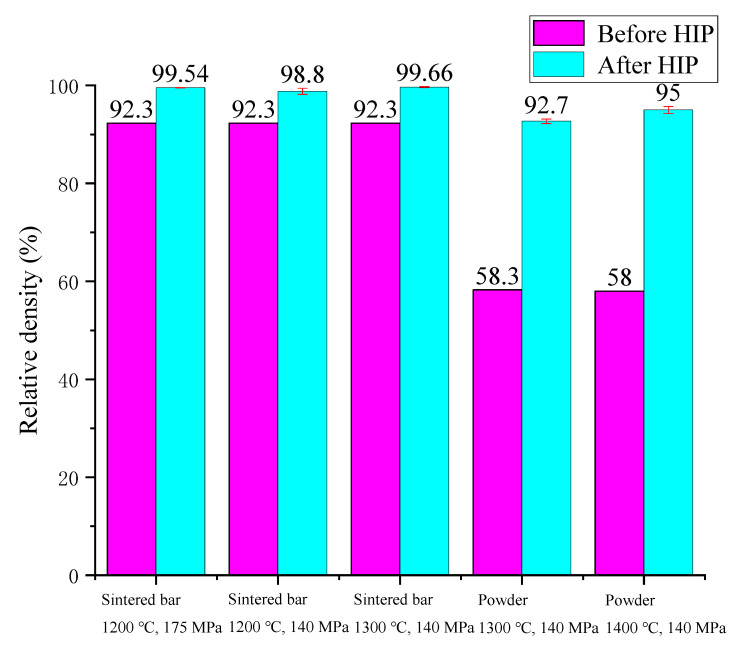
The relative density of tungsten alloy before and after HIP.

**Figure 6 materials-15-08647-f006:**
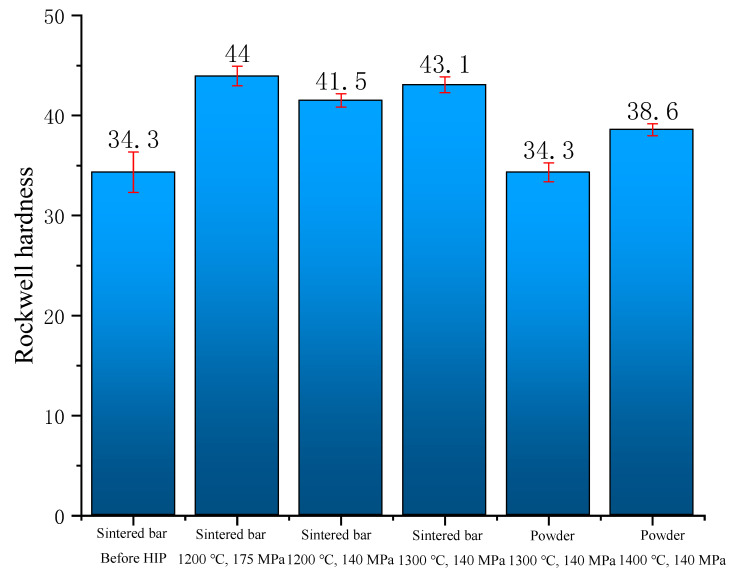
Rockwell hardness comparison before and after HIP.

**Figure 7 materials-15-08647-f007:**
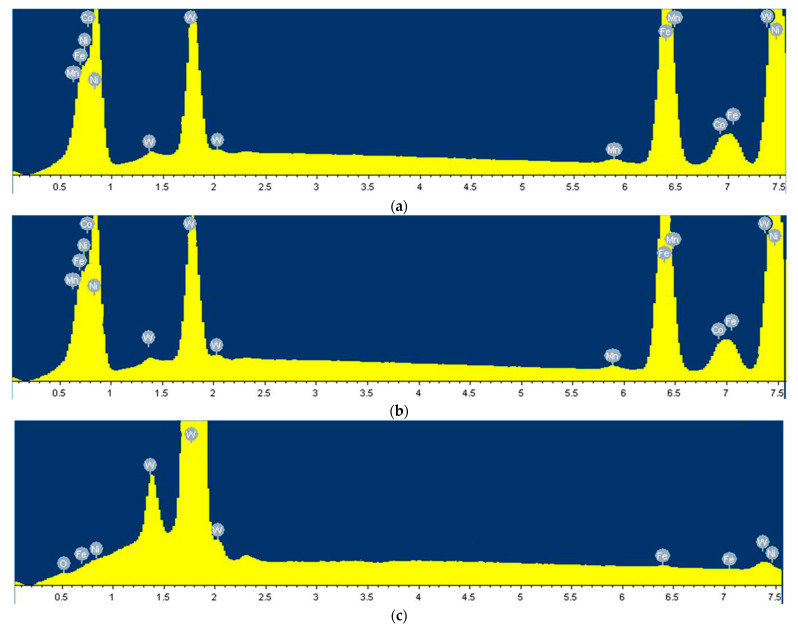
The chemical composition and element distribution of the sintered bar strengthened by HIP. (**a**) The chemical composition of bonding phase at 1200 °C and 140 MPa. (**b**) The chemical composition of bonding phase at 1300 °C and 140 MPa. (**c**) The chemical composition of tungsten particles at 1300 °C and 140 MPa.

**Figure 8 materials-15-08647-f008:**
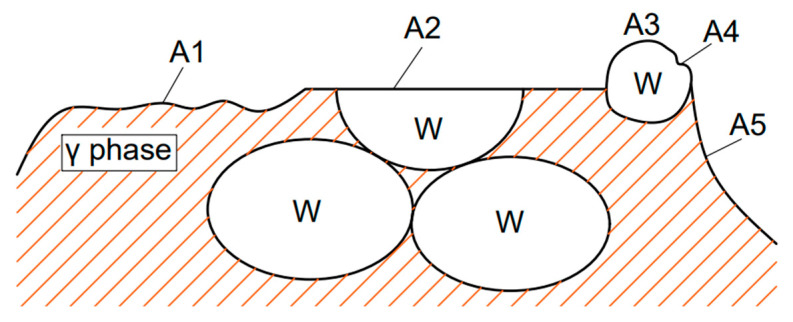
Fracture modes of tungsten alloys [19].

**Figure 9 materials-15-08647-f009:**
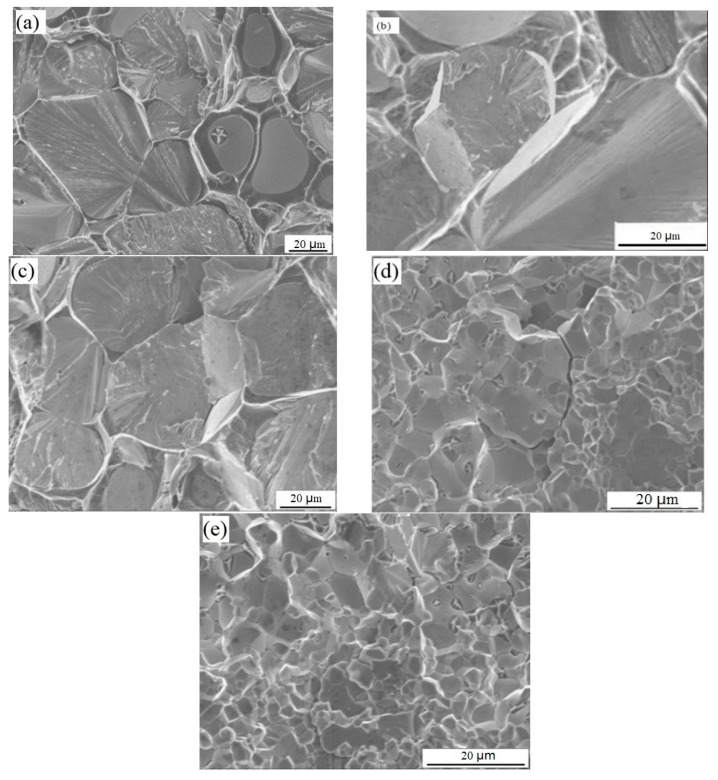
Fracture morphology of tungsten alloy under different HIP parameters: (**a**) sintered bar: 1200 °C, 175 MPa; (**b**) sintered bar: 1200 °C, 140 MPa; (**c**) sintered bar: 1300 °C, 140 MPa; (**d**) powders: 1300 °C, 140 MPa [19]; (**e**) powders: 1400 °C, 140 MPa [19].

**Table 1 materials-15-08647-t001:** Chemical composition of 93W-Ni-Fe heavy alloy (wt%).

Raw Powder	Mass Fraction/%	Purity/%	Major Impurity/%	
			O	C
W	93	99.98	<0.01	<0.005
Ni	4.45	99.98	<0.012	<0.005
Fe	2.2	99.98	<0.01	<0.006
Co	0.3	99.95	<0.035	<0.01
Mn	0.05	97.75	<1.55	<0.15

**Table 2 materials-15-08647-t002:** The main parameters of the HIP equipment.

Type	Size of Forming/mm	Maximum Temperature/°C	Maximum Pressure/MPa
RD-450	∅450×900	1600	200

**Table 3 materials-15-08647-t003:** The process parameters of HIP.

Material	Temperature (°C)	Pressure (MPa)	Dwell Time (h)
Sintered bar	1200	175	4
Sintered bar	1200	140	4
Sintered bar	1300	140	4
Powder	1300	140	4
Powder	1400	140	4

**Table 4 materials-15-08647-t004:** The corresponding percentages of each element in Figure 7.

Corresponding Picture	Element	Category
(a)	Mn(0.47) Fe(25.64) Co(4.04) Ni(48.27) W(18.87)	weight percent
	Mn(0.58) Fe(30.76) Co(4.59) Ni(55.09) W(6.88)	atomic percent
(b)	Mn(0.42) Fe(24.20) Co(3.81) Ni(45.73) W(20.91)	weight percent
	Mn(0.52) Fe(29.31) Co(4.40) Ni(52.72) W(8.98)	atomic percent
(c)	O(0.76) Fe(0.20) Ni(0.27) W(94.34)	weight percent
	O(7.68) Fe(0.57) Ni(0.75) W(83.03)	atomic percent

**Table 5 materials-15-08647-t005:** Comparison of mechanical properties under different process parameters.

Parameter		Material	Elongation/%			Section Shrinkage/%		
			Original	HIP	Improvement	Original	HIP	Improvement
1200 °C, 175 MPa		sintered bar	13.7	17.9	30.7	23.1	30.1	30.3
1200 °C, 140 MPa		sintered bar	13.7	16.5	20.4	23.1	29.3	26.8
1300 °C, 140 MPa		sintered bar	13.7	20.1	46.7	23.1	33.2	43.7
1300 °C, 140 MPa		powder	13.7	9.3	−32.1	23.1	13.1	−43.3
1400 °C, 140 MPa		powder	13.7	11.4	−16.8	23.1	15.7	−32.3
$\sigma_{b}$//MPa			$\sigma_{p0.2}$/MPa			Fracture Strength/MPa		
Original	HIP	Improvement	Original	HIP	Improvement	Original	HIP	Improvement
910	973.3	63.3	657	713.6	56.6	854.4	1392.4	538
910	942.2	32.2	657	698.9	41.9	854.4	1334.2	479.8
910	1057.3	147.3	657	765.2	108.2	854.4	1582.8	728.4
910	793.6	−116.4	657	523.5	−133.5	854.4	913.2	58.8
910	876.5	−33.5	657	641.1	−15.9	854.4	1039.7	185.3

## Data Availability

Not applicable.

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
