# Peer review of "Effect of Hot Isostatic Pressing Process Parameters on Properties and Fracture Behavior of Tungsten Alloy Powders and Sintered Bars"

_materials, 2022, doi:10.3390/ma15238647_

Round 1
Reviewer 1 Report
Necessary explanations are specified in the PDF file and in the relevant sections.

Reviewer 2 Report
The work is devoted to the study of the effect of HIP temperature and pressure on the properties and behavior during destruction of 93W-Ni-Fe heavy alloy powders and sintered bars. This trend is relevant. This trend is relevant. However, the work requires significant improvements due to shortcomings in the design of experiments and the description of the results.
Notes:
1. The authors significantly underestimate the density of existing tungsten alloys. The introduction presents a range of densities (16.5~9.0g/cm3), which is significantly lower than the values ​​presented in modern reviews (up to 19.3 g/cm3)[Recent Progress in Processing of Tungsten Heavy Alloys], [Effect of Rare Earth Metals (Y , La) and Refractory Metals (Mo, Ta, Re) to Improve the Mechanical Properties of W–Ni–Fe Alloy–A Review].
2. The paper does not indicate the theoretical density, which the authors use to calculate the real densities of alloys. The density values ​​need to be corrected and the results recalculated if necessary.
3. The method of preliminary sintering of samples (temperature, pressure and time) is not clear. This method (Argon arc welding) is rarely used for sample sintering in powder metallurgy. The use of this method for sintering heavy alloys is not presented in modern reviews. It is not clear differenc in density, hardness and strength of the samples obtained by this method and modern analogs. Therefore, the samples obtained by this method cannot be used as the only object for comparison. It is necessary to additionally compare all obtained properties (density, hardness, strength, elongation, fracture mode) with samples sintered by modern methods (liquid-phase sintering, SPS, microwave sintering, etc.).
4. The selected sintering modes differ in several parameters at the same time. At a pressure of 175 MPa, only one batch was sintered at a temperature of 1200°C. All other samples were sintered at a different temperature and pressure. Therefore, the conclusion about the influence of pressure is not correct.
5. The influence of the HIP temperature during sintering of the samples is represented by only two values ​​(1150 and 1300). The influence of the HIP temperature during powder sintering is represented by other values ​​(1300 and 1400). Conclusions about the effect of temperature for different workpieces and different temperatures are of little scientific value. The influence of temperatures and pressures must be presented in the form of dependencies.
6. Conclusions that At 1300°C and 140MPa , the sintered bar to achieve the best mechanical properties are limited by experimental conditions. Without comparison with analogues obtained under other sintering conditions, it cannot be concluded that these sintering conditions provide the best properties. It is necessary to compare with analogs and/or supplement the experiment to find a complete dependence.
7. The dependence of Rockwell hardness on the measurement number does not make sense. It is necessary to present the dependence of hardness on the density of samples. It is also necessary to compare the obtained hardness values ​​with analogues.
8. Elongation, ??//MPa, ??0.2/MPa, fracture strength should be presented as a function of density or other parameters.
9. The reference to the source [19] in the caption to Figure 7 must be removed. In the description of Figure 7, it is not clear which samples were used for sintering (bar or powder).
10. The term fracture strength must be used instead of fracture stress.
Conclusion
In the work, the influence of pressure and temperature of sintering on the properties of the alloy was found. Probably, an increase in the sintering temperature and pressure leads to a decrease in porosity, which leads to an increase in the strength, hardness, and plasticity of the tungsten alloy. However, randomly selected sintering modes do not allow one to unambiguously determine the influence of the main HIP parameters (temperature, pressure, workpiece type). The use of an unusual sintering method (arc melting) without an analysis of the main parameters and the lack of comparison of the obtained results with the existing values ​​of density, strength, hardness and plasticity do not allow us to confirm the conclusions drawn. The work requires significant improvements to be recommended for acceptance.
Reviewer 3 Report
The paper is devoted to investigation of the effect of the hot isostatic pressing (HIP) regimes on the density, microstructure parameters, and mechanical properties of heavy tungsten alloy 93W-Ni-Fe.
There are some critical remarks to the paper contents, which don’t allow Reviewer to evaluate this work positively.
1. First, it is worth noting that this paper overlaps partly with the work by the same authors published in Materials Today Communications journal (https://doi.org/10.1016/j.mtcomm.2022.103576) (This is reference [19] in the new paper submitted for review to Materials). In the new paper submitted to Materials, there are some results published earlier. In Reviewer’s opinion, the worst moment is that the level of details of the experimental procedure in the new paper is much lower than in the previous one. This makes the analysis of most results obtained impossible.
2. The authors described the sample preparation procedure very poorly. Much more details on the liquid-phase sintering regimes as well as detailed description of the results of investigations of the commercial alloy are necessary. Usually, there is no such low density (92.3%) in the W-Ni-Fe alloys if the sintering process is conducted properly. The possibility to achieve a high relative density and a uniform distribution of the g-phase inside the sintering material is always an advantage of the liquid phase sintering method. As a consequence, the authors obtained very poor mechanical properties of commercial samples obtained by conventional method. Here detailed comprehensive comments are necessary. The authors should expand “Materials and methods” section essentially – describe the procedure of mixing the powders, provide the information on the oxygen and carbon contents in the powders, describe the mixing regimes, provide the information on the equipment, justify the choice of HIP regimes, etc.
3. The authors should provide the information how the theoretical density of the tungsten alloys was calculated. Tungsten is known to dissolve in the g-phase easily during the liquid sintering. Because of the above, actual densities of the tungsten alloys are always smaller considerably than the theoretical value calculated in the assumption of additive contributions.
4. The authors wrote in “Materials and methods” section that the initial density of the powders in the steel capsules was close to 0.6. In Fig. 3, the density of the initial samples equal to 58.0-58.3% is given. How did the authors determine this quantity? If there are open pores in the samples, it is impossible to measure their density by Archimedes method.
5. The authors should compare the results of investigations of the mechanical properties obtained to the ones of commercial alloy 93W-Ni-Fe.
6. The capsules from medium carbon steel used by the authors can react chemically with the tungsten alloy. The W-Fe and W-C interactions are the most dangerous since these ones lead to the formation of brittle phases. In particular, because of the above, the carbonization of the surface layers takes place often when making the tungsten alloys by SPS in graphite molds. For this reason, the authors should provide the information on the microstructure of the central and surface parts of the samples.
7. The authors should provide the information on the microstructure of the sintered samples and, that is especially important, on the character of distribution and chemical composition of the g-phase in the sintered samples.
8. One can see in Fig. 3 that the cylinders processed by HIP at 1400 оС have the maximum densities. The densities of the cylindrical samples processed by HIP at 1200 and 1300 оС were smaller. Fig. 4 demonstrated the reverse results contrary – the values of hardness for the samples processed by HIP at 1200 and 1300 оC were higher than the one of the sample processes at 1400 оC. This contradiction is not discussed in the work.
9. The tension curve presented in Fig.9 is not a true one. It is an engineering curve obtained directly from the hardware of the testing machine. Since the tension curve presented has an abnormal form, not typical for the tungsten alloys, the authors should present the curves for all samples as well as comment the unusual multistage forms of these ones. In Reviewer’s opinion, the tests were conducted without the use of the strain gauges and, hence, one cannot talk about correct determining of the yield strength and elongation to failure. These characteristics can be determined correctly in the tests with the strain gauges attached to the sample surfaces only.
11. The cracks on the surfaces of the samples of high strength tungsten alloys are observed at the hardness measurements often. Please provide respective information or present the photographs of the imprints from the indenter.
12. The dependence of the hardness on the time of processing by HIP for the initial sample is presented in Fig. 4. It is not a suitable format for the data presentation since the initial samples were not processed by HIP. The authors should revise the data presentation format in Fig. 4. Also, the authors should show the uncertainties of determining the Rockwell hardness in Fig. 4.
13. One can see in Fig. 7 that the microstructure of the samples obtained by processing of cylindrical workpieces differs drastically from the one of the samples obtained by HIP of the powders. In the cylindrical samples, the strong tungsten grains are surrounded by plastic g-phase whereas in the powder samples such microstructure with plastic interphase boundaries wasn’t formed. Note that the tungsten particles in the powder samples have faceted shapes that is typical enough for the case of solid phase sintering. This, in particular, is seen well on the fractures of the powder sample (HIP, 1400 oC, 140 MPa) (Fig. 7e). However, in Fig. 10 in [19] for the same powder sample (HIP, 1400 oC, 140 MPa), the microstructure typical for the liquid phase sintering is shown. Such a mismatch of the results is surprising.
14. In conclusion, the authors should formulate clearly the scientific novelty of the results obtained. The fact that HIP leads to the increase in the density is well known, there is nothing new in this result. The HIP process is a low productive one and is not applicable for the mass production of the tungsten rods. Consequently, the authors should formulate and state the scientific novelty of the results.

Round 2
Reviewer 2 Report
The authors have done considerable work and eliminated several significant comments. Now the work has become clear and new comments have appeared. There are also some shortcomings in the work, which also need to be eliminated.
Remark 1
In modern studies, the maximum density of alloys (19.3 g/cm3) is greater than that indicated in the review (16.5~19.0g/cm3). In addition, the authors calculated a theoretical density value (18.595g/cm3), which also turned out to be lower than the value reported in other studies. The theoretical density value, which the authors calculated from known density values ​​for the chemical composition used, may differ from the actual density of an alloy with a relative density of 100%. It is necessary to use the known value of the density of the alloy of the same composition from another article, or to make an additional analysis of porosity by another method (metallographic) and present the results in the figure. This question is important because the increase in density is one of the main results of the application of HIP.
Remark 2
The description of the Archimedes drainage method can be shortened as it is detailed in ASTM B 962-08.
Remark 3
The text of the article alternates between sections describing structures and properties: Density (characterizes the structure), hardness (property), EDX (composition), mechanical properties (properties), fracture behavior (microstructure and properties).
Such an order of sections does not allow a complete description of the composition-structure-properties relationship. Hardness is described without taking into account data on composition and microstructure. Strength is described without microstructure data. It is necessary to arrange the sections in the order corresponding to the modern paradigm of materials science: composition - structure properties. The description needs to be corrected. Conclusions must also be presented in the specified order.
Remarks 4
The size of the letters in the figures (fig. 1, fig. 4, fig. 5, fig. 6) is too small. It is necessary to increase the letters.
Remark 5
Cu, Sr, Sr, Dy and other "formed" elements must be removed from the chemical composition (Figure 7., Table 4) . The formation of new elements during HIP is impossible. This error needs to be corrected.
Remark 6
The authors did not compare the obtained values ​​of density, strength, hardness with the values ​​from other studies of tungsten alloys. It is necessary to add samples from other studies to the table and make a comparative analysis.
Remark 7
The text describes the diffusion of tungsten particles during sintering (line 156). Diffusion of W particles is very slow at the HIP temperature. The sintered particles W are too large for diffusion. Diffusion of W atoms into particles is possible. It is necessary to explain what particle diffusion means or correct the text. Text should be corrected.
Remark 8
Numerical values ​​of the main results are required: increase in strength, elongation, change in area and ductility of the obtained material.
Reviewer 3 Report
No remarcs
